# Semantic Icons: A Sentiment Analysis as a Contribution to Sustainable Tourism

**Juan Pablo Vázquez Loaiza [1],\*, Antonio Pérez-Torres [2] and Karol Marylin Díaz Contreras [3]**

1   Departamento de Ingeniería Agroforestal de la ETSIAAB, Universidad Politécnica de Madrid,
    28040 Madrid, Spain
2   Grupo de Investigación de la Gestión de las Mipymes, Universidad Politécnica Salesiana,
    Cuenca 010102, Ecuador; jperezt@ups.edu.ec
3   Salesian Language Institute, Universidad Politécnica Salesiana, Cuenca 010102, Ecuador; kdiaz@ups.edu.ec
\*   Correspondence: jp.vazquez@alumnos.upm.es

**Abstract:** The construction of this research was developed to reflect the way information and communication technologies (ICT) have transformed the tourist distribution channel. This phenomenon has caused, in the first place, the direct interaction between tourists and tourism operators and, second, the appearance of real virtual intermediation actors, a situation that disadvantages the activity of traditional travel agencies and causes immersions in reintermediation processes at risk of compromising its permanence in the market. On the other hand, in a sustainable tourism context, this work represents an opportunity for intermediation agencies in terms of a value management practice as they can develop sustainable promotion processes that promote, for example, the protection of the territory, the conservation of resources, and cultural rescue. Therefore, according to travel agencies, to directly influence the distribution chain, we verified this work to demonstrate the use of digital language as a benefit in the design of sustainable tourism products. Thus, from the methodological discipline of discourse analysis, we created sets of words with semantic content that were valued through the sentiment scales of the Facebook social media network. The results showed that digital promotion favors airline companies and hostels more than a sustainable tourism environment as such. Finally, from the study of probability and density equations, predictive models were used to configure linguistic icons in promoting sustainable tourism.

**Keywords:** semantic analysis; ICT; essential marketing; tourism and sustainability; value and tourism

## 1. Introduction

The gradual growth of tourism can be represented by an annual average of 4% during the last eight years, which represents a displacement of over 393 million people [1], indicating an opportunity for impulsive tourist activities. It can also favor the progress of countries as it also provides positive impacts to their economy. However, it can also present negative repercussions regarding the protection of the environment due to the development and abrasion of their resources [2,3]. This issue, seen from the perspective of sustainable tourism, must be managed together with the generation of employment and wealth; the conservation of culture and patrimony, social inclusion, and respect to diversity; the efficiency of resources; and the fight against poverty [4].

A great number of researchers have addressed these problems by motivating and proposing the promotion of tourist destinations by distinguishing different aspects. For example, the synergy between faith and religion as a cultural essence for the delineation of strategies [5–7]; the awareness of animal protection that regards them as elements that improve tourism and the economic potential of a place [8–11]; the handcrafting and agricultural impulse [12,13], along with ancestral practices [14,15]

as a revitalizing economic alternative in rural areas; the development of gastronomy entrepreneurships as culture value; and the fight against poverty [16,17].

However, according to an ontological dimension, a tourist destination has to be studied beyond these aspects, and even include psychological and emotional factors due to the fact that sustainable tourism is seen as the result of development, is understood as a response of the actors' needs—travelers and the local community—and their long term projection capacity [18]. All these factors are a necessity for the adoption of marketing strategies [19]. Furthermore, this approach can be consolidated with an approximation of evaluation through economic, environmental, socio-cultural, institutional, and infrastructure indicators [20–22]. Although, according to all stakeholders, there exist some other factors that must be analyzed, but have been minimized, such as the phenomenon of 'overtourism', where some democratic terms need to be integrated into the rights of people [23].

In this research, we describe the theoretical bases and orientations of interest as we try to create a model of social responsibility. Besides, we examine the necessity of generating a conscience in travelers toward respect for the territory, as intermediation agencies are responsible for the promotion of activities both inside and outside its boundaries [24]. Additionally, in the value chain model, the distribution of tourist offerings constitutes a facilitator between tourism operators and tourists, and becomes an important link to spread, drive, and ensure the sustainability of tourist destinations [25]. Furthermore, it has met with resilient arguments to face de-intermediation in the development of social responsibility [26].

The aims are to sustain the structure of tourism intermediation [27] and adapt to the transformations produced by the interference of information and communication technologies (ICT), that some researchers [28,29] have addressed, such as:

1. The increase of the operators and their capacity to establish a vertical and direct relationship with the tourist, thus avoiding intermediation;
2. The necessity of personalization processes undertaken by the intermediaries, thus recovering power and position in the channel;
3. The effectiveness of the construction of products that respond to the tourists' needs; and
4. A transformation in the activity of the final consumer, who demands the implementation of co-creation processes.

Additionally, to maintain sustainability, an evaluation must be made for corporative management to ensure satisfaction among all the interested parties, resource management [30], the reduction of the process, and the integration of the micro, small, and medium businesses to the quality objectives [31] such as quality of service, cost reduction, consistency, and information security.

Thus, to exploit the use of ICT, and give value to the management of tourism intermediation, this research was constructed as a follow-up study of previous work. This defined a technological ecosystem, where the models of business-to-business and business-to-consumers would work on a dynamic of intelligence and internal management based on interaction and knowledge [31]. The intention is that, from an innovation attempt, a logical sequential flow of identification, capture, creation, and diffusion of knowledge that contemplates touristic horizons regarding accessibility, accommodation, attractions, and services is adapted [32]. In addition, instituting it simultaneously as a tool for the value chain to facilitate the management and integration of the actors [33].

It must also be extended not only as a simple solution of optimization but as a contribution to gathering relevant information for decision-making and the definition of strategies. For instance, the information regarding the determination of content associated with travelers and their relationship with the existing attractions within the touristic space [34]. Thus, the aim is to try not to reach a critical status where the traveler wastes their traveling experience by seeing themselves embedded by the excessive use of technology, obtaining, in this way, a false perception and satisfaction, and causing isolation of the individual or becoming a case of e-lienation [35] due to loss of authenticity [36–38] and the intra-personal authenticity destroyed by the technophilia [39].

As a practice of social manifestation, consequences are spread through social networks, becoming significant input for marketing studies and merchandising planning, and should be managed through the adoption of technology [40]. In this research, social and environmental variables were an important alternative contribution within the 'triple bottom line' (Guidance framework for measuring business objectives that considers economic, environmental and social aspects [41]) [42]. In this way, sustainability is not only provided by the business, but also accomplishes different precepts of the administration and preservation of resources for future generations, attending to the need for sustainability for the development of society [42].

Hence, the actions that the business must perform through social networks must consider an implicit benefit for the travelers; in addition, the interaction and immediacy of response can be formalized as they are features that affect the brand attitude and corporate confidence [43]. Furthermore, they are configured in hedonist forms for tourists and associated to use the channel and income for the business [44]. In addition, they indicate that in the context of tourism, a large volume of information prevails that is dependent on effective communication [45]. In this research, acculturation manifestations are represented [46] as derived from the social impact exteriorized through stress factors, adaptation processes, and negative emotions [47,48] as well as integration and polarization facilitators, which transform the society and create new consumption opportunities [49].

Therefore, to discover and understand the meaning of the digital social expressions that go beyond simple words, this research is attached to the horizon of discourse analysis (DA), as the comprehension of the social dictions of natural language and their meanings are promoters of a real construction of the society [50,51]. Moreover, as described in [52], it helps to establish a logical inferential structure for accessing knowledge as a contribution to Aristotle´s logic, so that language analysis explains the nature of the environment and facilitates the detection of requirements [53]. Arguments of special interest for this work try to offer an opportunity of applied investigation and practical contribution to innovation in intermediary tourism agencies.

Likewise, the importance of lexicon and discourse, according to [54], favors facing the control that it exercises on any society, as demonstrated in the investigation conducted in [55]. This fact was intended to be transformed with this investigation, ensuring that control favors intermediary agencies as a conciliating entity between supply and demand. It is then important to delve into the semantics to discover the relationships that, from the inspection of the 'corpus', facilitate the configuration of new statements of values adapted to a reality [56], that are elaborated and guided by cognitive structures [57] that define the linguistic signs as the representation of a tourist's object of interest. Therefore, to benefit the results in this research, and according to [58], which acknowledges the contribution of DA to the analysis of conflicts in economic, political, anthropological, and psychological environments, it is expected to be the pragmatic basis of a new approach into the paradigm of sustainable tourism.

Additionally, it is important to discover if digital social expressions represent, in some measure, the desires, beliefs, and values of a society and the tourist environment. They must respond, from a cognitive line to an enforceable action from prior planning that understands that the textual and contextual structure allows for the formulation of the message in the receptor, giving them a fast and convenient meaning [59]. However, their subjectivity must be registered in the subjectivity of the social context—understood as the tourist environment in this study—and discourse [60]. All of this is to recognize the 'systems of dispositions', which are latent in the environment and the social structure [61], so that it can reaffirm itself and cause a transformation, benefiting sustainable development if it is necessary.

However, why is it important to analyze the text in tourist publications? Because answers are given, mainly the ones regarding the studied theories by [62], who highlighted that: (a) Language is constituted as a graphic system that expresses the social behavior and its knowledge as semiology [63]; (b) through linguistic registers, the variations of expression are evidenced, so prediction is manifested; and (c) semantics contain the meanings of social context. For this reason, and in agreement with [64], these signs represent the relationship or influence of an object—that is expected to be the tourist



environment—and an interpreter, who is a traveler or tourist. However, as suggested by [65], they naturally express physical phenomena, for example, a sunset or a dance, or unconscious human behaviors that, by the way, will serve to make sense of something that is not consciously conceived [66].

Finally, this research will serve to help tourism intermediaries manage sustainable development. Although it acknowledges the contributions that can be obtained through marketing practices, it is not oriented to the centralization of common management that considers the necessities of the visitors versus the products/services to motivate purchasing and consumption processes per se [67,68]. Nor does it express an analysis of the environment, and through the harnessing of power, the public relations and alliances with suppliers [69], but strives to reach sustainable development such as the one underpinned in the essential marketing paradigm that interprets the behavior and meanings associated with codes and social archetypes. This is so that a perceptual concept can be constituted by the physical concept and the imaginary concept, which comprehends cultural diversity, ethnic groups, geography, and socio-economy [70] to such an extent that symbolic goods, whose commercial value develops an understanding of their cultural values, can be defined [71].

In summary, because social networks are a communication trend and interaction with them promotes social transformation, the purpose of this research was to understand the reality that travel agencies must provide to improve the sustainable development of tourism. This requires the promotion of tourist destinations as a contribution to the value chain. Moreover, due to symbolic interactionism, where individual behavior is outlined over the meaning of things [72], it might be significant to promote a proposal that can sensitize the tourist toward an emotional connection with the territory and the country that they visit.

Thus, after the application of discourse analysis techniques, semantic analysis, and the analysis of emotions, this work tried to prove if the travel agencies promoted sustainable development practices, which, from their meaning, were of interest to the tourists. However, even if this research could identify linguistic icons associated with the tourist context, they would be constituted as a commercialized success of the offer, especially for airlines and accommodation providers. This finding motivates the continuation of further investigation that reviews the reality of the benefits of a sustainable environment.

## 2. Fundamental Theories of Research and Sustainable Tourism

This section supports theories that connect by shaping an approach toward sustainable tourism from the management of intermediation in the exploitation of digital content. In this sense, first, it is emphasized that thanks to the incursion of ICT, the traditional model of tourist intermediation has been transformed since agencies and operators have witnessed the arrival of specialist distribution agents through digital channels [29]. In addition, operators have found opportunities for direct negotiation with travelers, dispensing with the services of intermediaries. This reality has forced intermediaries to outline strategies to strengthen and compete in a way that favors their permanent place in the market [73,74].

Therefore, by trying to facilitate new management models through digital technological adaptation, the research adheres to the value theory as it allows for the design, definition, and delivery of products as an intention of competitiveness [75] and differentiation, that is also conceived in a generation of awareness for the protection of the territory [24,25]. However, it must also contribute to the intelligent integration of the actors in the distribution chain, which involves the generation and exchange of information and knowledge [26,27,33,76]. In addition, it should help improve the tourist experience as an essential activity [77] by also encouraging the protection of resources, the rescue of culture, and sources of employment [2–17].

Therefore, if virtual channels are considered as the scenario in which the intermediation action takes place, it is appropriate to assume that the digital content should act as an input entity for the behavior analysis, as the information delivery of a sustainable message that, once again, improves travel practice and promotes the development of the tourist destination.

Consequently, this purpose was carried out through the practice of discourse analysis. This is mainly because digital channels, and mainly virtual social networks, encourage the exchange of information that demonstrates the behavior and expression of society. Manifestations through which it was sought to establish the basis for intelligence processes that, thanks to the theory of sets, could denote relationships between axiomatic elements [78] consisting of linguistic components and an assessment understood from the analysis of feelings.

Derivations that can ultimately be inferred in concept from the paradigm approach of digital marketing, while provoking a practice in such a way that the negotiator can build a sufficiently striking message, are products of the co-creation of knowledge [44,49]. It not only seeks to only close a sale, but to offer something that causes a truly pleasant feeling in the consumer as a new cultural reality by highlighting the characteristics of the object [70,71] that, in this case, are semantically constituted in the cultural, natural, and social properties and values of the tourist environment [50–53,59–66].

## 3. Materials and Methods

### 3.1. Discursive Methodological Configuration

The discursive analysis of the text in this research was laid out through a scheme suggested by [79] that is illustrated in Figure 1. It represents sustainable tourism as a conceptual concept supported by the essential marketing and value theory as key concepts for development. Furthermore, for the application of the method, discourse analysis is the main discipline in an approach to social semiotics. Finally, the own structures of the language will be analyzed in distinct to the components of the grammatical phrases.

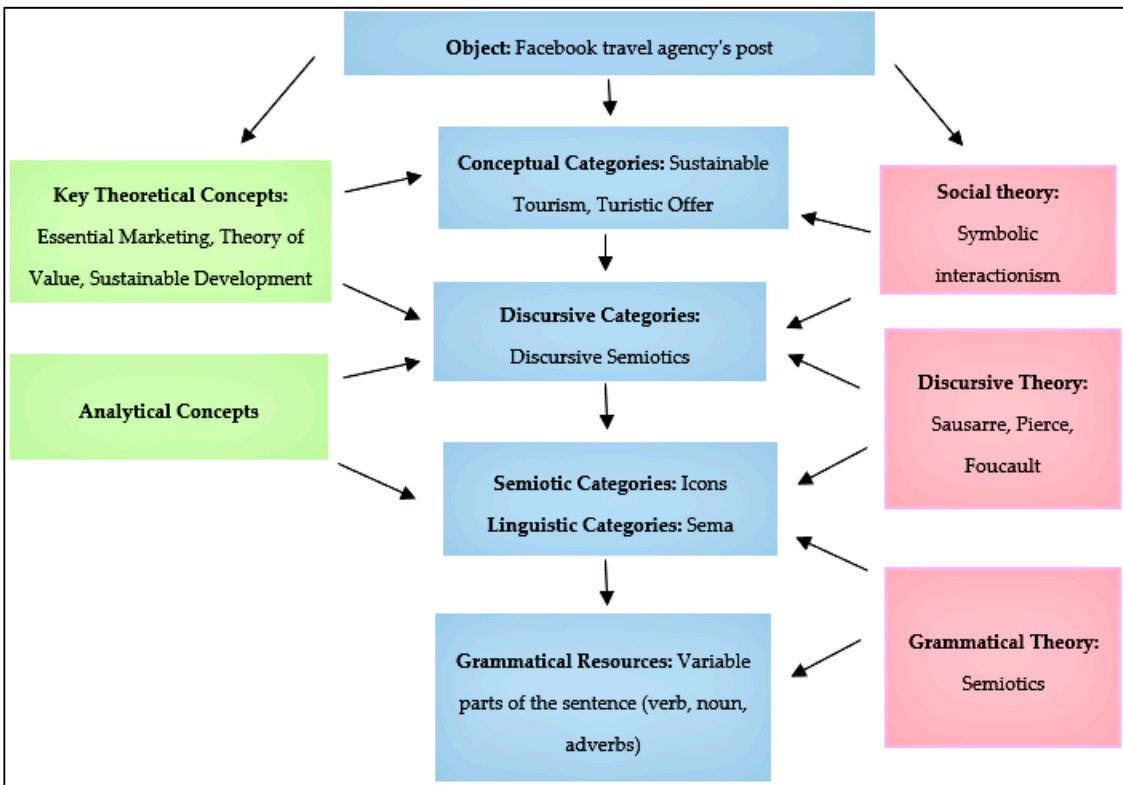

**Figure 1.** Discursive methodological configuration.

### 3.2. Discursive Analysis

The logic of discursive analysis was directed through procedures supported by the computing linguistics of (1) the object of study, for example, posts by Facebook travel agencies, as input data;

(2) the morphology, semantics, and pragmatic knowledge as actions of the computing processing; and (3) the predictive icons of the language as output text. This process was adapted from [80] with the following phases:

A computing morphological analysis that:

- Automatically extracts posts made by travel agencies on Facebook. To do this, NCapture was used, which is a web browser extension of the Nvivo software. From here, the universe of the words to be studied was created.
- From the universe, words were classified by language, numeric characters, hashtags, and another called 'trash' that corresponded to those that did not provide semantic information such as symbols and email addresses as well as those with a length of one character as they did not represent semantic content.
- Produced a second classification that distinguished customer names, street names, prices expressed in letters, web addresses, and words like 'cell', 'cel', or 'tel' as they only refer to telephone information provided for contact, and consequently, they do not affect the semantic content.
- Spelling mistakes were corrected, mainly the ones regarding the use of tildes (e.g. the word 'acomodacion' (accommodation) was changed to 'acomodación'. Additionally, some compound words were identified and unified (e.g., the words 'Abu' and 'Dabi', which appeared separated were put together, resulting in 'Abu Dabi'.)
- A group of words that were called 'irrelevant' was recognized. Here, the grammatical prepositions (e.g., 'por' (by), 'para' [for], 'entre' (between)), articles, and conjunctions (e.g., 'pero' (but), 'porque' (because)) were included.
- Concluded with the conformation of elemental grammatical units through the free online lemmatizer developed by the Group of Data Structures and Computing Linguistics of the University of Las Palmas (http://www.gedlc.ulpgc.es/investigacion/scogeme02/lematiza.htm).

A semantic analysis that from the posts:

- Coded the meaning of the words, without distinction of the association of the tourist context, through the Nvivo software. Here, the nouns, adjectives, verbs, and adverbs were identified. Thus, what this research intended to do was to represent the tangible elements associated with the perception of the senses with nouns, quality of the feelings toward the nouns with adjectives, the exercise of action with verbs, and the conditions and circumstances linked with the ones above-mentioned with adverbs.
- Conducted a morphosyntactic analysis of these words, according to the different contexts of a sentence in the case of homonymy and polysemy through the Stilus (https://www.mystilus.com) software.

A pragmatic analysis that:

- Related the semantic content of the tourist environment through the associations of nodes as shown in Figure 2.
- This analysis used the Chi-Square ($X^2$) statistical test with a significance level of $\alpha = 0.05$, where the hypothesis to be tested was the association between the grammatical categories and the nodes of the tourist context if the p-value was superior to the critical value.

An analysis of feelings was made to validate the lexicon of touristic words in order to undertake a simulation on how much these words could be attractive to tourists, understanding that, according to [60], an expression of feeling coming from an individual impulses or impacts on the feelings of a collectivity; therefore, it is important to identify the positive words in different posts with the purpose of provoking positive semantic impacts. To accomplish that expressed above:

- Within the facilities of a travel agency, clients were observed who requested information motivated by undertaking an intention to travel. Of these, 20 volunteers of legal age, regardless of gender,

purchasing power, or interest in tourism products, digitally evaluated the words of the lexicon. To do this, they used the sentiment scale that Facebook configures to assess the publications and resembles a 6-point Likert scale [81,82]. For instance, 1 = Angry 😡 ; 2 = Sad 😢 ; 3 = Amazed 😮 ; 4 = Like 👍 ; 5 = I enjoy 😆 ; and 6 = I love ❤️ .

- To achieve a level of concordance, Kendall's W statistical test (Equation (1)) was applied and calculated using Statistical Package for the Social Science (SPSS) software for a value of W acceptance close to 1.

$$W = \frac{12 \sum_{i=1}^{k} S_i^2 - 3n^2 K(K+1)^2}{n^2 K^2 (K-1)} \tag{1}$$

where *K* is the number of characteristics subjected to evaluation; *N* is the number of elements of study; and *S* is the addition of the marks obtained in each of the evaluated characteristics.

- The calculation algorithm found the best combination of judges for an acceptable value minimum of $\alpha = 0.7$. The results were obtained by calculating the highest value between combinations from 2,3,4,5 ... *n* evaluators.

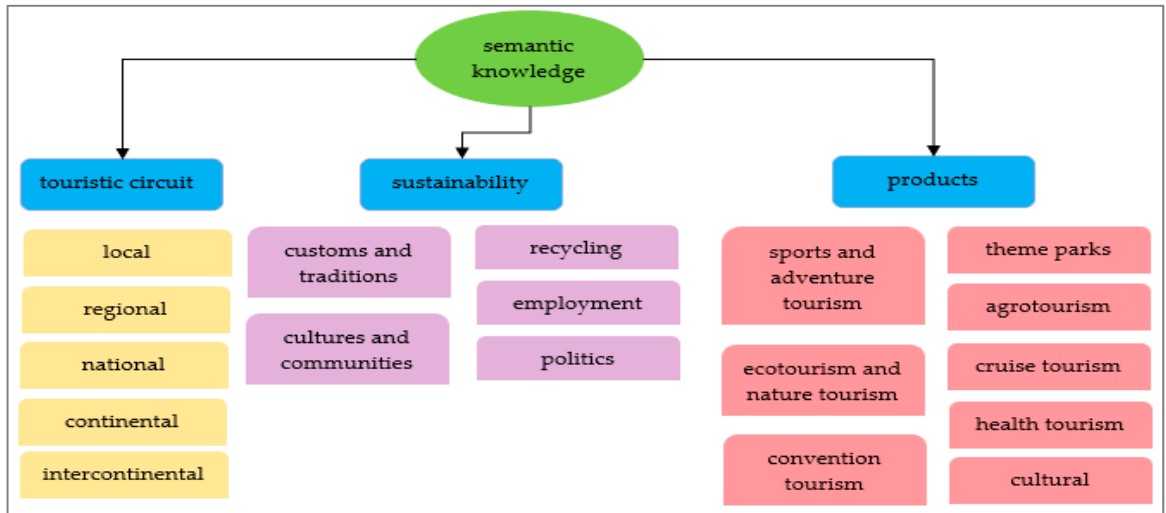

**Figure 2.** Pragmatic knowledge representation. Adapted from PLANDETUR 2020 (Design of the Strategic Development Plan of Sustainable Tourism for Ecuador. https://www.turismo.gob.ec/wp-content/uploads/downloads/2013/02/PLANDETUR-2020.pdf).

*3.3. Predictive Analysis*

As a complement oriented to analysis intelligence, this research intended to discover the linguistic icons from the lexicon. This, according to [83], defines the horizon of the marketing expansion by means of placing each word into different ranks that follow the vital cycle of marketing via the knowledge of (a) the novelty, (b) the trend, (c) the new fashion, (d) the top, (e) the consolidation, and (f) the obsolescence. To do this, the probability thresholds considered are shown in Table 1.

In this sense, the normality of the frequencies was first studied through the Kolmogorov–Smirnov (K–S) test, with a significance level $\alpha = 0.05$. In this way, if the p-value obtained for the categories was higher than the significance level, the normality of the distribution would be accepted. Subsequently, the function of the density and the definite integral (Equation (2)) was undertaken with the study of the goodness-of-fit applied through the coefficient of determination $R^2$ to represent the behavior of the linguistic expressions and test the forecast model of each word set.

$$P_w \left( a \leq X \leq b \right) = \int_a^b f(x) d(x) = F(b) - F(a) \tag{2}$$

where $P_w$ is the probability of the word; *a* is the lower limit of the area (1 for all cases); and *b* is the upper limit of the area, which is the frequency value of the word.

Therefore, from the density function, by integrating the lower limits of 1 and upper limit equal to the frequency value of the word, the space in which each word would be located according to the horizon of market expansion could be determined later. The purpose for which, through Chebyshev's Theorem (Equation (3)) to find the probability of occurrence of an event for any random variable *X* as a function of *k* standard deviations concerning the arithmetic mean $\mu$, is detailed in [84].

$$P(u - k\sigma < X < u - k\sigma) \geq 1 - \frac{1}{k^2} \tag{3}$$

where *K* is the to the threshold of distribution of the area under the curve meeting $-3 \leq k \leq 3$; $\mu$ is the arithmetic mean; $\sigma$ is the standard deviation; and *X* is the random variable word such $X \in S$.

**Table 1.** The threshold of probability for marketing expansion.

| Status | Area From | Area Up |
| --- | --- | --- |
| The novelty | −3k | −2k |
| The trend | −2k | −1k |
| The new fashion | −1k | μ |
| The top | μ | 1k |
| The consolidation | 1k | 2k |
| The obsolescence | 2k | 3k |

*3.4. Study Case*

To carry out the study and application of the methodology, the city of Cuenca in Ecuador (2°53′51″ S, 79°00′16″ W) was taken as a study case, with the purpose of contributing to the improvement of tourist activities in a city listed as a Cultural Heritage of the World site, thanks to its architectural, archeological, and natural potential. This city has received a World Travel Award in 2017 as the lead travel destination for short holidays in South America. In addition, the consolidation of the city as a sustainable destination is sought.

Conversely, reasons to use intermediation have responded to the fact that during the last three years, according to what has been observed by the author in the given inventory by the Ministry of Tourism of Ecuador, the travel agencies, operators, and wholesalers have registered an increase regarding the cessation of business, a fact that is affecting the economic development of the city. For this reason, it is expected that the results obtained in this research could provide development pathways through the practice of smart technological alternatives. Likewise, according to previous diagnostic studies, it is evident that businesses in this area are not strong about their digital practices for the promotion of their different offers due to:

1. The quality of their websites, which are deficient, as seen from the promotional content and their functionality [85].
2. The lack of estimates regarding basic management oriented to optimization practices in different search engines [86].
3. A lack of knowledge about the administration of social networks [31].
4. An average e-readiness of three on a scale of 1 to 5. This fact confirms the above-mentioned, aside from the lack of interest in the starting processes oriented to the adoption of smart digital technologies [31,87].

From this reality and in concordance with the inventory of tourism intermediation businesses in 2018, 151 travel agencies were recognized. However, of the 86 travel agencies who undertook activities on Facebook, only 49 had an official fan page, which limited the automatic extraction process with NCapture, and 22 demonstrated the constant activity with a minimum of 100 monthly posts.

The number of entities was finally used for the analysis of their posts over a period between 1 January and 31 December 2018. During this period, travel agency administrators made 6057 posts, thus forming a total universe from which, with the Analysis Stats 2.0 software, a finite-sample was applied with a 95% confidence level and a margin of error of 1%. Finally, we determined a sample size of 3714 posts (D).

Thus, finally the sampling was conducted at convenience and only studied the companies with a total number of posts that exceeded 100 monthly, because at the discretion of the digital marketing experts [88,89], a frequency between two and five daily posts causes interactivity between the client and the company and substantially improves the return on investment. This added 5412 advertising discourses (X), which was 1968 more than the sample size, and corresponded to 22 businesses. Subsequently, to establish the percentage of the proportion of 69% between D and X, the number of posts to study for the business was determined, as shown in Table 2.

**Table 2.** Sampling of posts.

| Enterprise | Posts (Q) | Post by a Company (69%) |
|---|---|---|
| E1 | 589 | 404 |
| E2 | 340 | 233 |
| E3 | 110 | 75 |
| E 4 | 125 | 86 |
| E 5 | 329 | 226 |
| E 6 | 219 | 150 |
| E 7 | 384 | 264 |
| E 8 | 259 | 178 |
| E 9 | 127 | 87 |
| E 10 | 149 | 102 |
| E 11 | 157 | 108 |
| E 12 | 288 | 198 |
| E 13 | 125 | 86 |
| E 14 | 254 | 174 |
| E 15 | 245 | 168 |
| E 16 | 111 | 76 |
| E 17 | 281 | 193 |
| E 18 | 117 | 80 |
| E 19 | 163 | 112 |
| E 20 | 109 | 75 |
| E 21 | 677 | 465 |
| E 22 | 254 | 174 |

## 4. Results

After applying the methodology, the obtained results were as described below. These are mainly represented through the set theory to facilitate understanding.

*4.1. Morphological Analysis*

Some base sets were constructed for the classification of terms and to facilitate the morphological analysis, the results of which are presented in Figure 3, with some samples of the content of these sets shown in Figure 4.

**Set U** = {all the terms included in the posts}
**Set S** = {Spanish language words}
**Set E** = {English language words}
**Set I** = {Italian language words}
**Set F** = {French language words}
**Set A** = {Austrian language words}
**Set N** = {numeric characters chain}

**Set H** = {hashtags}

**Set T** = {trash terms}

**Set X** = {length 1-character terms}

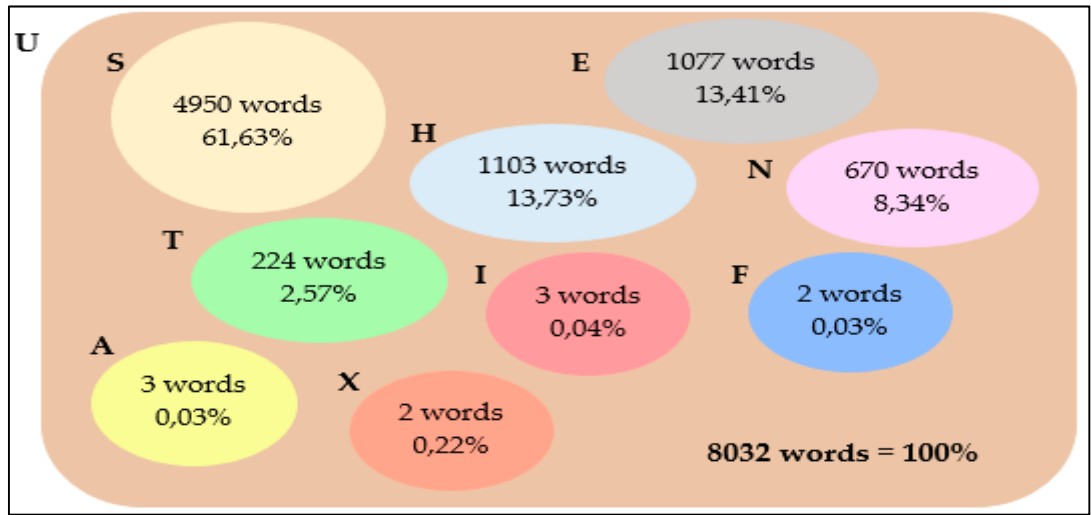

**Figure 3.** Word sets.

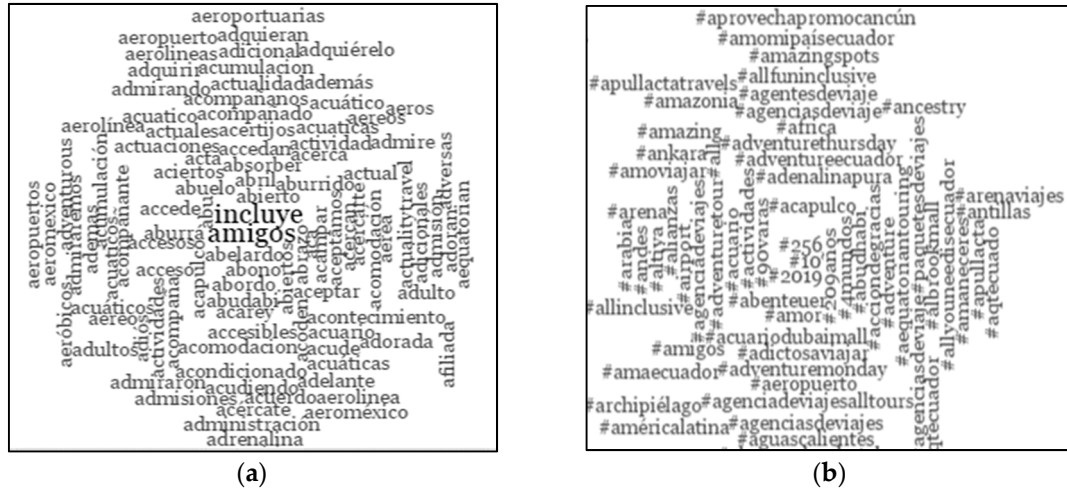

**Figure 4.** Subsets samples. Panel (**a**) shows the Spanish word set (S), and panel (**b**) shows the trash word set (T).

From the obtained data, the study focused on the Spanish word set because the setting of this research mainly expressed the information in this language and because the size of the information was mainly in Spanish. Therefore, from the Spanish set (S), new subsets were configured as shown in the following.

**S′** = (words that represent names of customers, names of streets, web addresses, and prices)

Some examples of words included in this set were: 'www. Teléfonos, Gabriel' (proper name referred to a customer and no to a character associated with the environment), 'Córdova' (derived from Presidente Córdova street).

**S″** = (words with corrected spelling mistakes)

Examples of this set were the words: Iguazu (it should be Iguazú), 'gastronomico' (it should be 'gastronómico').

**R** = (articles, prepositions, and grammatical conjunctions).

**S₂** = {independent words that form compounds of two words}

Examples of words included in this set are the words 'Abu' and 'Dabi', which appeared independently when the real word is 'Abu Dabi'.

$S_3$ = {independent words that would form a compound of three words}

Examples of words included in this set are the independent terms 'Amari', 'Havodda', and the 'Maldives' that would form the compound word 'Amari Havodda Maldives', and refers to a proper name of a place of accommodation.

$S_4$ = {independent words that would form a compound of four words}

Examples of words included in this set are the independent words 'Barceló', 'Maya', 'Grande', and 'Salinas', that would form the compound word 'Barceló Maya Grande Salinas', which also refers to a proper noun of a place of accommodation.

$S_5$ = {independent words that would form compound terms of five words}

Some examples of this set are the independent words 'Museo', 'de', 'Cera', 'Madame', 'Tussauds' that would form the word phrase 'Museo de Cera Madame Tussauds', which refers to an attraction in the United Kingdom.

$S_6$ = {Independent words that would form phrases of six terms}

Examples of words that were found in this set include 'Holiday', 'Inn', 'Express', 'Miami', 'Doral', 'Airport'. Written together, they make the phrase 'Holiday Inn Express Miami Doral Airport', which is also a place of accommodation.

S7 = {irrelevant words}

S2a = {compound words of two terms}

S3a = {compound words of three terms}

S4a = {compound words of four terms}

S5a = {compound phrases of five terms}

S6a = {compound phrases of six terms}

To count the number of words and obtain the total of terms for the subsets ($S_{2a}$), ($S_{3a}$), ($S_{4a}$), ($S_{5a}$), and ($S_{6a}$), Microsoft Excel was used as shown in Equation (4). In this way, the final presentation of the (S) set and its subsets can be illustrated in Figure 5:

$$= IF\ (LEN(TRIM(range\_cell)) = 0;0;\ LEN\ (TRIM(range\_cell)) - LEN\ (SUSTITUIR\ (range\_cell;\ "";"")))\ + 1 \tag{4}$$

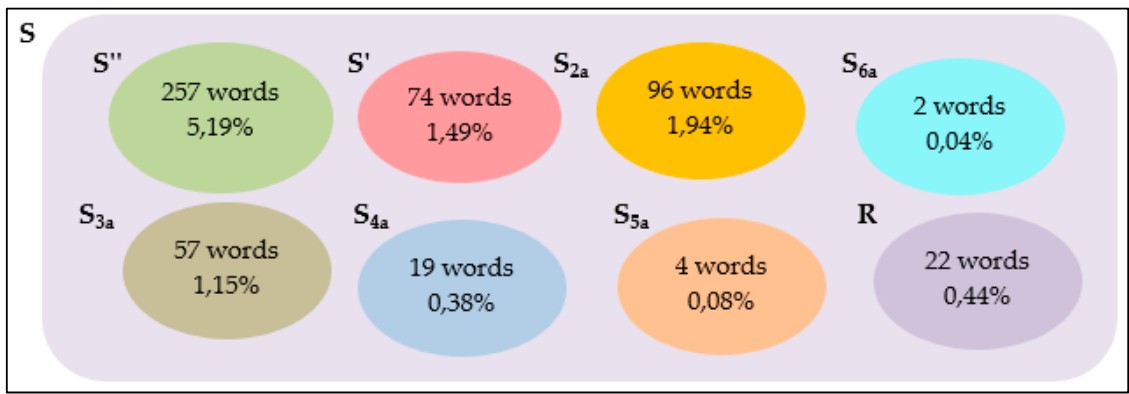

**Figure 5.** Spanish word subset distribution.

Thus, by operating the (S) set and all its subsets through Equation (5), the relative set (W) was built. Here, the lemmatization process was applied after identifying the canonical or lexeme forms, consolidating the last set, named Lexicon, with 1999 validated terms for the semantic content.

$$W = (S\backslash S'\backslash S''\backslash S_2\backslash S_3\backslash S_4\backslash S_5\backslash S_6\backslash S_7\backslash R) \cup S_{2a} \cup S_{3a} \cup S_{4a} \cup S_{5a} \cup S_{6a} \tag{5}$$

### 4.2. Semantic Analysis

In the Nvivo program, the words of the lexicon were coded according to four grammatical categories, whose distribution is shown in Table 3.

**Table 3.** Semantic grammatical distribution.

| Set | Number of Words | Percentage of the Lexicon |
|---|---|---|
| Adjectives | 296 | 15% |
| Adverbs | 39 | 2% |
| Nouns | 1396 | 70% |
| Verbs | 268 | 13% |

Additionally, the Stilus program was simultaneously used to avoid grammatical disambiguation and identify cases of polysemy and homonymy, as shown in Figure 6. The result would permit, not only the representation of a grammatical category but also the establishment of an adequate sense of the relationship with the other components of the sentence. Therefore, for instance, a term could be classified as an adjective, noun, or any verb form; however, when discovering their relationship in the 'corpus' of the sentence, it would be convenient to code it as a noun.

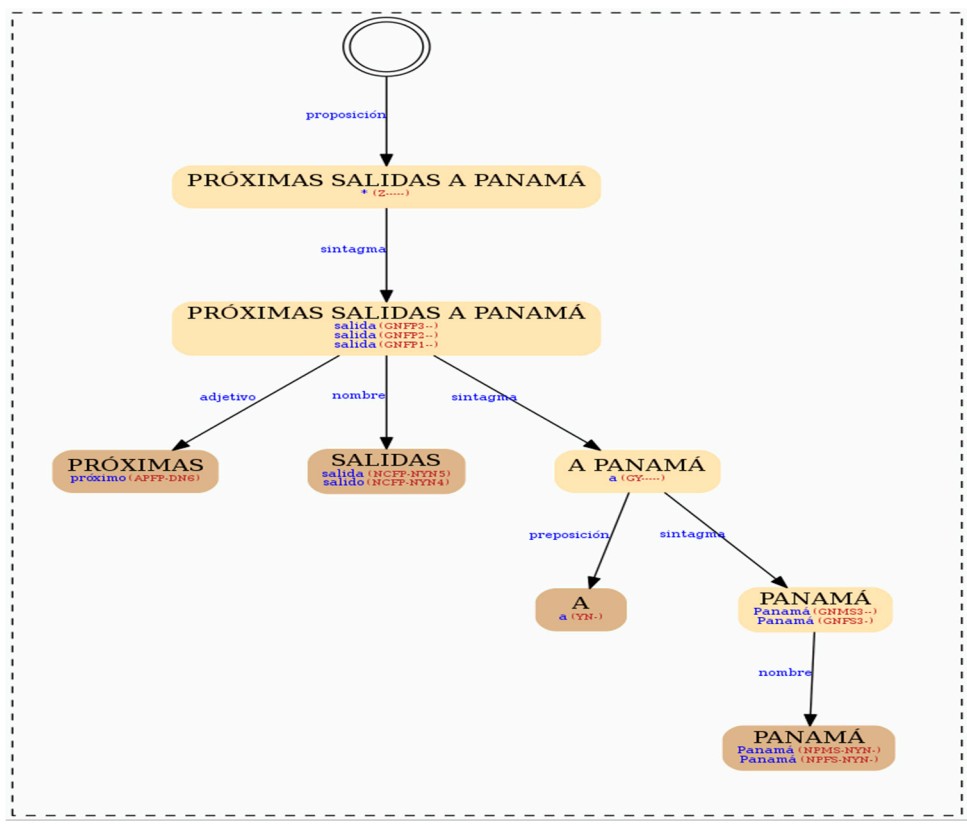

**Figure 6.** Example of grammatical disambiguation.

Finally, complementing the frequency distribution from the grammatical categories, the codification concludes by providing the repetition frequency per word, by following with Figure 7. These results refer to the apparent preference of the intermediaries to mainly promote air transportation services and hosting. This situation would not place sustainable tourism as the fundamental core and purpose because the words regarding values, identity, culture, and attractive spaces that the countries offer, showed a low repetition. For example, 'airport' was mentioned 840 times, in contrast

with 'Galapagos', which only appeared 118 times, 'happiness' appeared 60 times, 'culture' appeared 17 times, and 'archeology' or 'community', seven times.

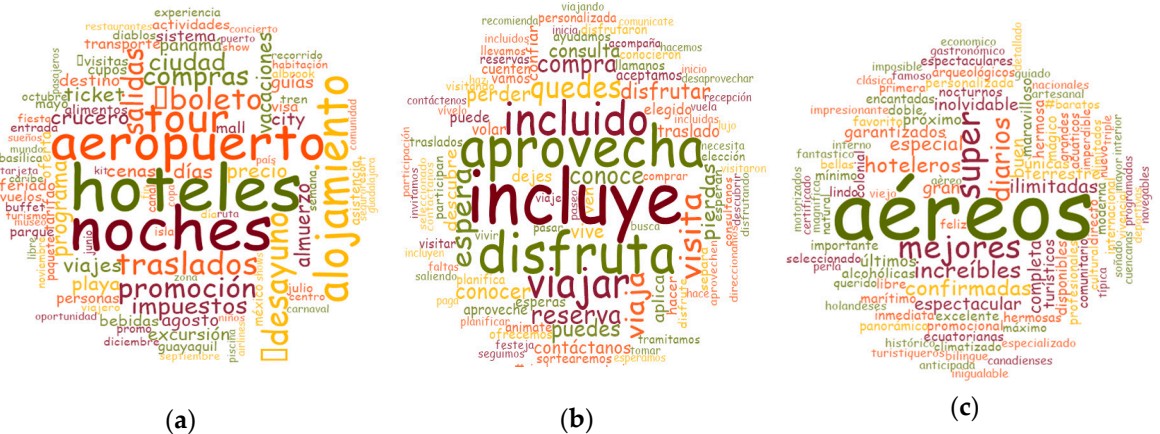

|  (**a**)  |  (**b**)  |  (**c**)  |

**Figure 7.** Words frequency. Panel (**a**) shows the nouns, panel (**b**) represents the verbs, and panel (**c**) illustrates the adjectives.

### 4.3. Pragmatic Analysis

To provide an additional sense to the *corpus*, the semantic content was compared with the tourist context. The results provided the grammatical categories that had the most influence in the interpretation of the message, as shown in Table 4.

**Table 4.** Relation of grammar sense versus touristic context.

| Tourism Context Variables | X² | df | Critical Value | Bilateral Asymptotic Significance | Excluded Categories |
|---|---|---|---|---|---|
| Sustainability | 26.69 | 16 | 26.2962 | 0.045 | Adverbs not relevant |
| Touristic circuit | 336.48 | 16 | 26.2962 | 0.000 | All grammatical categories and circuit values are relevant |
| Product | 796.85 | 40 | 55.7585 | 0.000 | Adverbs not relevant |
| Product: Ecotourism | 150.80 | 8 | 15.5073 | 0.000 | Natural Reserves not relevant |
| Product: Cultural | 60.96 | 16 | 26.2962 | 0.000 | Shamanism and adverbs not relevant |
| Product: Sports and adventure tourism | 10.08 | 2 | 5.9915 | 0.006 | Adjectives and adverbs not relevant |
| Product: Health tourism | 3.132 * | 6 | 12.5916 | 0.792 | All grammatical categories and health tourism are relevant |

(*) Relationship hypothesis rejected. Therefore, the grammatical content does not provide a relationship with the tourist context.

In this sense, the results are interesting because, from a semantic point of view, nouns are the words that provide a strong relationship. Nouns provoke a feeling in tourists to start or develop a travel experience since they indicate the positive orientation of the speaker [90]. Nevertheless, other words should be outlined for the rescue and promotion of each country's identity and values, and to guide it toward sustainability and development.

### 4.4. Sentiment Analysis

Once the grammatical relationship and the touristic content of the significant relationship were discovered, this research tried to find the words that, from a cognitive order, would provoke a greater satisfaction or pleasure in the tourist. Consequently, the sentiment evaluation granted sets of words that were more attractive in terms of:

Recognizing the criteria similarities toward the valuation that the judges would give to the nouns, adjectives, and verbs. When a p-value close to zero was obtained for all the categories, as shown in Table 5, it reflected non-significance for Kendall's W. Consequently, several iterations were performed until an acceptable coherence value was obtained, as shown in Tables 6–8. This was undertaken due to the wide number of variables, which might suggest subjectivity, thus impeding the ability to obtain an acceptable coherence value.

**Table 5.** Criteria similarity according to grammatical categories.

| Grammatical Category | Kendall's W Coefficient | # Judges |
| --- | --- | --- |
| Nouns | 0.133 | 20 |
| Verbs | 0.195 | 20 |
| Adjectives | 0.116 | 20 |

**Table 6.** Similarity coefficient for nouns.

| The Best Combination of Evaluators | Kendall's W Coefficient | # Judges |
| --- | --- | --- |
| E3–E7 | 0.522 | 2 |
| E3–E7–E17 | 0.534 | 3 |
| E3–E7–E17–E13 | 0.481 | 4 |
| E3–E7–E17–E13–E16 | 0.628 | 5 |
| E3–E7–E17–E13–E16–E18 | 0.690 | 6 |
| E3–E7–E17–E13–E16–E18–E6 | 0.663 | 7 |
| E3–E7–E17–E13–E16–E18–E6–E2 | 0.719 * | 8 |

(*) Combinations that meet the similarity criterion for $\alpha = 0.70$.

**Table 7.** Similarity coefficient for verbs.

| Best Combination of Evaluators | Kendall's W Coefficient | # Judges |
| --- | --- | --- |
| E4–E13 | 0.380 | 2 |
| E4–E13–E3 | 0.209 | 3 |
| E4–E13–E3–E6 | 0.423 | 4 |
| E4–E13–E3–E6–E11 | 0.537 | 5 |
| E4–E13–E3–E6–E11–E15 | 0.627 | 6 |
| E4–E13–E3–E6–E11–E15–E8 | 0.732 * | 7 |
| E4–E13–E3–E6–E11–E15–E8–E2 | 0.767 * | 8 |
| E4–E13–E3–E6–E11–E15–E8–E2–E9 | 0.802 * | 9 |

(*) Combinations that meet the similarity criterion for $\alpha = 0.70$.

**Table 8.** Similarity coefficient for adjectives.

| Best Combination of Evaluators | Kendall's W Coefficient | # Judges |
| --- | --- | --- |
| E7–E13 | 0.356 | 2 |
| E7–E13–E10 | 0.401 | 3 |
| E7–E13–E10–E15 | 0.497 | 4 |
| E7–E13–E10–E15–E18 | 0.586 | 5 |
| E7–E13–E10–E15–E18–E6 | 0.712 | 6 |
| E7–E13–E10–E15–E18–E6–E9 | 0.808 * | 7 |

(*) Combinations that meet the similarity criterion for $\alpha = 0.70$.

From the valuations, the methodological trial finally provided new subsets of words that were grouped according to the sentiments shown in Figure 8. These grant a significant derivation and might represent the linguistic icons that should be used to promote sustainable tourism. This miscellaneous constitutes an important input of analysis in which to plan strategic marketing. Similarly, travel agencies might strengthen their business through the promotion of transportation and accommodation.

Tourists might be attracted by the promotions if nouns were used such as *'love'*, *'happiness'*, and *'friendship'*; and adjectives like *'anthropological'*, *'ancestral'*, *'fun'*, *'exclusive'*, and *'delicious'*.

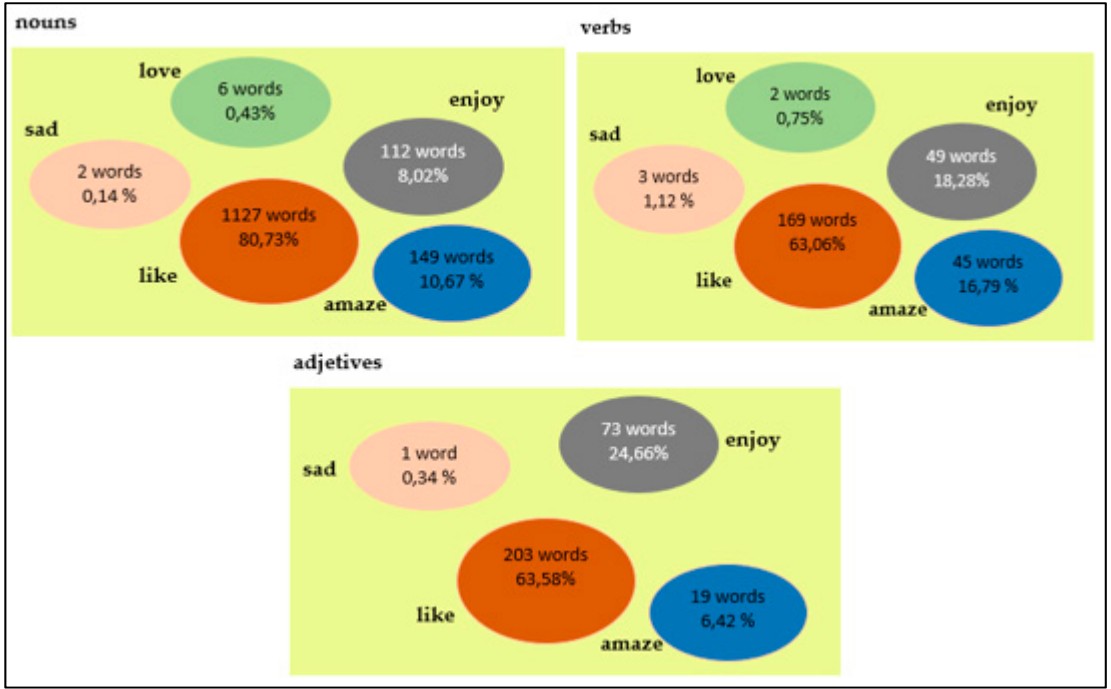

**Figure 8.** Set distributions according to the sentiment evaluation.

### 4.5. Predictive Analysis

The study conducted through K–S, as shown in Table 9, demonstrated that the frequency distribution for the grammatical categories was not adjusted to a homogenous reality with the significance level of $\alpha = 0.05$. This confirms the frequency polygons in Figures 9–11, which would justify the use of Chebyshev's Theorem because it is independent of the type of distribution that the variable has [82]. Therefore, to accomplish the predictive analysis, the density functions were built through the equation tests of tendency, and the results are shown in the mentioned figures that correspond to those functions where the best level of adjustment for the determination coefficient was obtained as $R^2$.

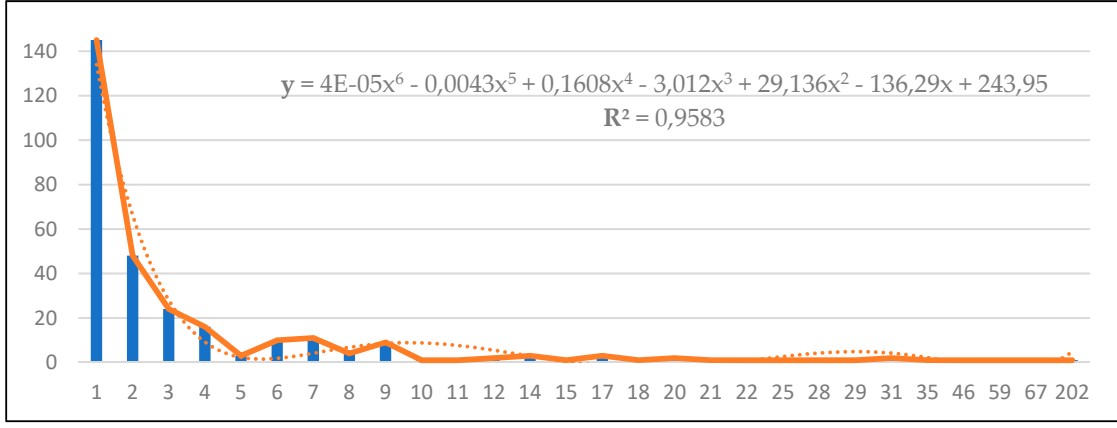

**Figure 9.** Adjective frequency distributions. Six-degree polynomial function.

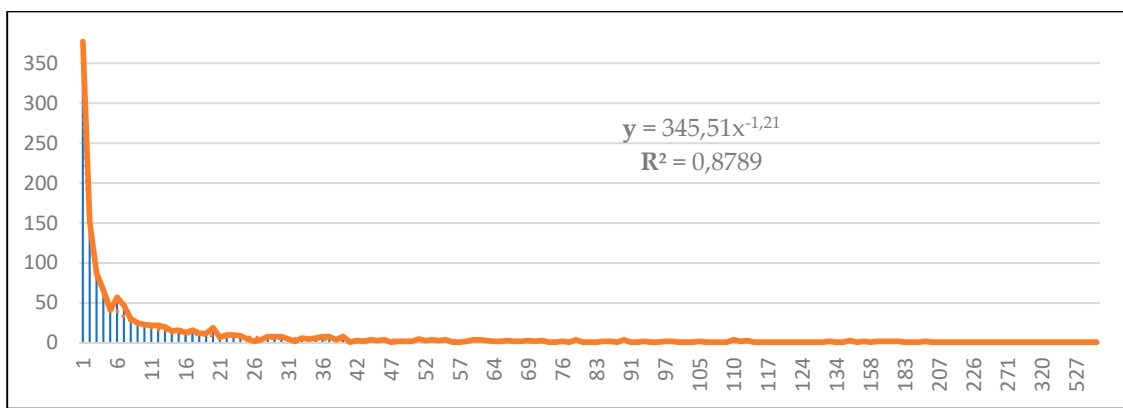

**Figure 10.** Noun frequency distributions. Potential function.

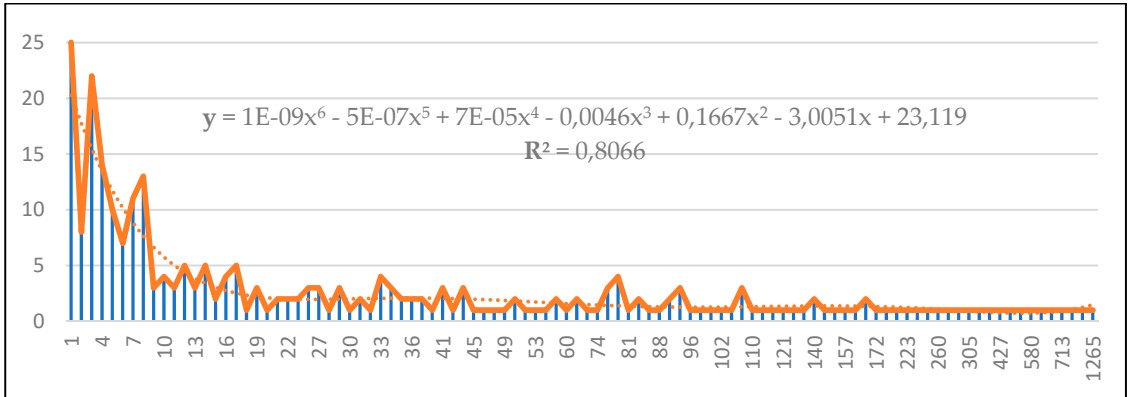

**Figure 11.** Verb frequency distributions. Six-degree polynomial function.

**Table 9.** Normality test.

| Categories | Z Kolmogorov–Smirnov | p-Value |
|---|---|---|
| Adjectives | 0.352 | 0.000 |
| Nouns | 0.376 | 0.000 |
| Verbs | 0.389 | 0.000 |

Where, after the determination of probabilities through density functions, they were obtained for the noun, adjective, and verb sets, and the position in which each word was placed on the marketing horizon. The results are exemplified in Table 10 and correspond to the terms that, from the analysis of feelings, received the highest assessment (love) that the tourist intermediary should try to reach the threshold, called 'the top'.

In this way, the usefulness of the study allows for the recognition of words both in feeling and threshold of probability that could benefit advertising practices. Thus, for example, it could be assumed that the combination 'Dream' + 'Enjoy' + 'Aquatic' + 'Ancestral', would attract the attention of tourists and possibly represent benefits for both the intermediary company and the territory as it derives an approximate publication with the value and focus of sustainable tourism.

On the other hand, after the application of the prediction and probability models, words were discovered, for example, that turned out to be located at a threshold of 'the novelty' and that casually receive the best evaluation of feelings. This means that, after the study, travel agencies can use words in their advertising messages that, while being attractive to customers, also promote sustainable tourism.

**Table 10.** Predictive analysis.

| Word | Set | Frequency | Probability Value | Probability Threshold | Status |
|------|-----|-----------|-------------------|-----------------------|--------|
| Feeding | Nouns | 19 | 0.61 | 2k | The consolidation |
| Lunch | Nouns | 147 | 0.86 | 3k | The obsolescence |
| Love | Nouns | 25 | 0.65 | 2k | The consolidation |
| Rainbow | Nouns | 1 | 0 | −1k | The new fashion |
| Happiness | Nouns | 1 | 0 | −1k | The new fashion |
| Friendship | Nouns | 2 | 0.8 | 3k | The obsolescence |
| Aircraft | Nouns | 9 | 0.49 | 2k | The consolidation |
| Amazon | Nouns | 10 | 0.51 | 2k | The consolidation |
| Handicraft | Nouns | 11 | 0.52 | 2k | The consolidation |
| Eat | Verbs | 1 | 0 | −1k | The new fashion |
| Dream | Verbs | 6 | $1.66611 \times 10^{-10}$ | −3k | The novelty |
| Enjoy | Verbs | 115 | $-1.59091 \times 10^{-8}$ | −3k | The novelty |
| Aquatic | Adjectives | 5 | $4.1770 \times 10^{-9}$ | −3k | The novelty |
| Aereo | Adjectives | 202 | 1 | 1k | The top |
| Ancestral | Adjectives | 7 | $4.26504 \times 10^{-9}$ | −3k | The novelty |

## 5. Discussion

The exploration and study of web content is a topic that contributes to intelligence, knowledge management, and business management. Technical areas might see this as advantageous because the representation of their reality tends to be objective. For social studies, this topic might be more complex because investigations such as the one presented here, base their experience on subjectivity and phenomenology analysis.

In this scenario, a previous semantic analysis conducted by [87] showed a contrast between supply and demand by using digital social network content, which facilitated the ability to attract travelers and promote the creation of new products to improve some brands. Therefore, to continue to manage value, this research complements the state of the art by mainly establishing an identification practice of the linguistic icons of social content inside the tourism context. This fact constitutes a base in which to go beyond the generation of Web 3.0 and Web 4.0 content, which were also of interest in this work.

Subsequently, an extraction and data analysis alternative was systematized through set theory and constructed from the discourse analysis theory. These theories contributed to codify and structure the information to forming a lexicon that represented the semantic value for sustainable tourism. It also sought to motivate the defense of the territory and cultural rescue precisely as part of that semantic value that, according to [91] and in accordance with the paradigm of essential marketing, advertising content favors a positive reaction to the enjoyment of travel and respect for the cultural, ethnic, geographical, and socio-economic diversity of the environment [92]. The contributions of this work, with the construction of signs that transform the traveler's psychology, have allowed for the generation of an interactive culture that provides apparent, circumstantial, and manifest evidence [70] that can be given, afterward, to the social–cultural system of the receptor. The findings of this work constitute part of the acculturation process [93] of social networks and in the paradigm of sustainability [94,95].

From a pragmatic point of view, it is evident that the advertising intentions of the studied intermediation agencies do not seem to be aligned with the purpose of sustainable tourism. The reality indicates, through linguistic signs in the sets of words, that the intermediaries prefer to contribute to building a better relationship with places of accommodation and transportation than with the country or territory. The semantic relationship did not observe a close connection with social identity and another paradigm that [93] acknowledged as city-brand, where the values and attractions are stimulated toward new development schemes.

Finally, focusing on the future, this investigation will continue with the development of a monitoring system, which will allow us to measure the evolution of the words within the scheme of the marketing life cycle. Additionally, it will be necessary to develop a study to prove that,

indeed, the semantics of development and sustainable tourism favors the commercial activity of the intermediary companies. What this study expects is to start a configuration of artificial analytics based on qualitative data mining, whose main objective is to structure deterministic and probabilistic rules that, through the application of structural equation models (SEM) and expert stochastic models, favor not only the construction of algorithms and automated platforms for the extraction and analysis of the data, but also for the reply of the information.

**Author Contributions:** Conceptualization, J.P.V.L.; Formal analysis, J.P.V.L., A.P.-T.; Methodology, J.P.V.L. and A.P.-T.; Investigation, J.P.V.L.; Resources, J.P.V.L.; Software, J.P.V.L.; Data Curation, A.P.-T.; Writing-original draft preparation, J.P.V.L., K.M.D.C.; Translation, K.M.D.C.

**Funding:** This research received no external funding.

**Acknowledgments:** We would like to thank the Group of Data Structures and Computational Linguistics of the University of Las Palmas for providing extended access to the use of the online lemmatizer as the quantity of information exceeded the number of allowable maximum transactions meant that we required a special connection to finish this research.

**Conflicts of Interest:** The authors declare no conflicts of interest.

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
