# Peer review of "Semantic Icons: A Sentiment Analysis as a Contribution to Sustainable Tourism"

_sustainability, doi:10.3390/su11174655_

Round 1

Reviewer 1 Report

- interesting paper, innovative approach 

- good English (minor check)

- extensive literature review but better presentation of cited aspects could be Applied (table?)

- in abstract - better explanation of goal, methodlology and contribution 

- better explanation of conducted Research should be provided (methodology - sample, time)

Author Response

Very grateful for your cordial and important contributions and suggestions.

Reviewer 2 Report

The research is quite interesting, I would like to ask the authors if they can improve the paper explaining why they have chosen a Likert scale of 6-points responses and not 5- or 7- (if they can provide literature supporting their decision). I would appreciate also if they could explain where they collected the 20 volunteer evaluators.

Congratulations for your research

Author Response

(The authors gave the same response as above.)

Reviewer 3 Report

The paper is very interesting given the fact that it tries to contribute to the discussion on the role/influence of social media in tourist behaviour. Authors try to provide insights  of what it is discussed on facebook pages of tourist agents and try to make sense of the posts. The paper has some great parts and some others that need re-structuring.

To my taste the Introduction is not good. It seams that the paper was written by two different people. The Introduction is not connected well to the rest of the paper (see handmade comments in scanned pages). I would suggest that 2 nes chapters are written: An introduction that gives an actual intro to the topic researched and a second chapter that gives us the Theoretical framework through which the data make sense and contribute to the current scholarly discussion. I don;t want to impose my opinion but the theoretical part might move around the Sentiment Analysis as a framework especially through the lens of Sustainability. To this comment i might add that one of the major drawbacks of this paper is the loose connection to  sustainability. I did not see how in fact the findings of the sentiment analysis of the posts connect to the concept of sustainability. Some good parts to make use for the  introduction are already there (see page 8/22) and should be placed in the introduction giving the problem and the problematic/need/importance of the research.

Another point for clarification for the Theoretical Framework is the use of many different theories (see comment for Set Theory,

Authors must also try and keep it simple in terms of different concepts: if we read the introduction we see a great number of concepts introduced BUT not analysed in the findings nor in the discussion: poverty, faith, democracy, rights, overtourism, Aristotle, Socio-Economy...My recommendation is to focus to only the working concepts within the data analysed.

2. Materials & Methods: this is the best part of the paper. Despite the fact that i am not an expert on computer sciences..authors seam to know exactly what they are doing and it is well presented to the reader albeit some minor comments (see written comments in scanned pdf) . It would be also very interesting to add the profile of the volunteers evaluators and support from a methodological point of view why and how they were selected. I only have a question on the 6point Likert scale instead of the 7point or 5point (this needs supporting reference). Finally, it wasn't clear to me -maybe i didn't see it- what type/kind of businesses are those involved in the research (hotels, others?) and how did you come up with the size sample?  please add this information also.

3. The data analyses also is a very strong chapter (some parts i have to be honest-do not fall within my field of expertise but it was rather clear). BUT still i didn't find the connection to Sustainability

4. Discussion: this is rather weak as a chapter. Mainly because we see many concepts introduced here for the first time instead of being introduced earlier. for instance "business intelligence", Web 2.0 which are very relevant but should have been part of the introduction or the theoretical framework and here they should have been connected to the  results. This section of the paper needs to be re-written aiming at the contribution of the results to sustainability.

Author Response

(The authors gave the same response as above.)

Reviewer 4 Report

This work has a lot of potential and was a refreshing addition. Semiotic research is unfortunately unusual in the tourism literature. It may have been the translation but the beginning of the document and therefore the purpose of the study was difficult to contextualize. Therefore, the merit of the study was muddled. I am sure with some work these valuable findings would be more widely understood and useful to readers of this important topic.

Author Response

In response, to the suggestions, I have rewritten the abstract mainly and some sections throughout the document trying to improve the use of language and comprehension.

Very grateful for your cordial and important contributions and suggestions.

Round 2

Reviewer 3 Report

point 1. i am still not convinced that quality of life is the same concept with good feeling and happiness. This needs to be reconsidered 

point 2. i totally appreciate the reponse . you should put it  exactly as you wrote it in the text "On the other hand..."

point 13. put icons in the text - is perfect in order to understand it 

Introduction. still -to my taste- needs major revision: starting paragraph is not clear (does it refer to global circumstances, what does displacement means..)

the fight against poverty is a huge concept and it is mentioned twice in the introduction and not at all embedded in the results onwards. i would suggest if authors want to deal with such huge concepts they have to better analyze and conceptualize their results accordingly 

gender references to both him/her should be carefully written 

english should be improved (use a native speaker) 

table 1 with theories is quite superficial as it is placed there . i still think it is better to devote a small chapter to theoretical concepts -well described- instead of a table AND fully develop the connection to sustain9state of the art) and what the research has to contribute to the concept of sustainability ability there  and then in the last section make comparisons of what exists  

table 1 : Set theory is also missing 

Reviewer 4 Report

There is still major editing for mechanics, grammar, writing style, word choice, and sentence structure needed. 

Content development lacks a comprehensive literature review. The theoretical framework is presented as a glossary as one table. This is not an acceptable foundation from which to develop a conceptual framework nor does it illustrate the study's contribution to the literature.

This submission changed very little the initial one. 

Author Response

Please see the attachment. Thank you all your recommendations.

Round 3

Reviewer 4 Report

No comments at this time. Grammar and mechanics still require editing. However, I have no other content or organization comments.

Author Response

A new grammar was made. In fact, some expressions with problem were found. Once again, very grateful for the right observations and contribute to improve the quality of the article